# LoRaWAN Based Indoor Localization Using Random Neural Networks

**Winfred Ingabire [1,2], Hadi Larijani [1,*], Ryan M. Gibson [1] and Ayyaz-UI-Haq Qureshi [1]**

1 School of Computing, Engineering and Built Environment, Glasgow Caledonian University, Glasgow G40BA, UK; winfred.ingabire@gcu.ac.uk (W.I.); ryan.gibson@gcu.ac.uk (R.M.G.); ayyaz.qureshi@gcu.ac.uk (A.-U.-H.Q.)
2 Department of Electrical and Electronics Engineering, College of Science and Technology, University of Rwanda, Kigali P.O. Box 4285, Rwanda
* Correspondence: H.Larijani@gcu.ac.uk

**Abstract:** Global Positioning Systems (GPS) are frequently used as a potential solution for localization applications. However, GPS does not work indoors due to a lack of direct Line-of-Sight (LOS) satellite signals received from the End Device (ED) due to thick solid materials blocking the ultra-high frequency signals. Furthermore, fingerprint localization using Received Signal Strength Indicator (RSSI) values is typical for localization in indoor environments. Therefore, this paper develops a low-power intelligent localization system for indoor environments using Long-Range Wide-Area Networks (LoRaWAN) RSSI values with Random Neural Networks (RNN). The proposed localization system demonstrates 98.5% improvement in average localization error compared to related studies with a minimum average localization error of 0.12 m in the Line-of-Sight (LOS). The obtained results confirm LoRaWAN-RNN-based localization systems suitable for indoor environments in LOS applied in big sports halls, hospital wards, shopping malls, airports, and many more with the highest accuracy of 99.52%. Furthermore, a minimum average localization error of 13.94 m was obtained in the Non-Line-of-Sight (NLOS) scenario, and this result is appropriate for the management and control of vehicles in indoor car parks, industries, or any other fleet in a pre-defined area in the NLOS with the highest accuracy of 44.24%.

**Keywords:** IoT; LoRaWAN; RSSI; indoor localization; RNN

## 1. Introduction

Localization is vital in developing smart cities applications with the Internet of Things (IoT) where ED can be monitored or tracked in both outdoor and indoor environments. GPS is broadly considered for outdoor localization applications [1]. Nonetheless, GPS is not suitable for indoor localization services due to solid and thick obstacles found in buildings, and a higher power consumption [2]. Furthermore, different wireless technologies such as Bluetooth [3], WiFi [4], and Zigbee [5] have been used to develop indoor localization systems but are only limited to the range of 30 to 100 m. Therefore, LoRaWAN, an emerging low-power long-range technology, has mainly been considered in the recent IoT advancements, including localization as a perfect solution in developing low-power wide-area IoT systems. Long-Range (LoRa) technology has been frequently used in developing energy-efficient and cost-effective localization systems for outdoor environments using RNN and achieved significant results [6]. However, the performance of LoRa based localization systems in indoor environments using RNN is yet to be explored. Indoor localization systems are essential for security, safety, and efficiency in buildings or working environments where they are also crucial for tracking patients and the elderly, especially for quick intervention in emergency or maintenance services.

Furthermore, LoRaWAN RSSI-based fingerprint indoor localization systems have been developed by different researchers using different fingerprint matching algorithms [7].

However, to the best of our knowledge, none have used the RNN algorithm in developing LoRaWAN RSSI-based indoor localization systems. This study uses RNN in developing a long-range power efficient positioning system applying LoRaWAN signal strength values to determine unknown X and Y position coordinates. The proposed system achieves higher accuracy than conventional positioning systems in current related studies and achieves an accuracy of 99.52% in LOS and 44.24% in NLOS.

The following are the key contributions of this work:

- The novelty of RNN approach with the current literature.
- Developing a novel LoRaWAN RSSI-based indoor localization system.
- Different RNN-based indoor positioning models are trained and tested applying different numbers of hidden neurons.
- Training and testing different RNN-based indoor localization systems with various learning rates.
- A comparative performance analysis of the obtained results with other results in the existing related work.

This paper is organised as follows: Section 2 briefly describes LoRa and LoRaWAN, Section 3 explores the related research studies. Section 4 presents the methodology implemented. Section 5 discusses the results obtained and gives a comparative performance analysis. Lastly, Section 6 gives the conclusions and future work.

## 2. Overview of LoRaWAN

LoRaWAN defines the protocol stack LoRa on the media access control layer. In addition, LoRa is a Low-Power, Wide-Area Network (LPWAN) technology designed specifically for IoT applications. It is an open specification managed by Semtech, which uses chip spread spectrum modulation and is usable at 868 MHz in Europe, 915 MHz in the USA, and 433 MHz in Asia. Figure 1 shows the key components of a LoRaWAN network connected in a star topology [8]. One or multiple gateways can receive information from several ED and forward the received messages to a network server. Each received message is allocated metadata information that is used for positioning in LoRaWAN networks. Therefore, LoRaWAN gateways are used as anchor points to determine the position of the ED. Furthermore, fingerprint localization in LoRaWAN networks uses RSSI values transmitted by the end ED to determine their locations. When more gateways receive the same information, the accuracy of the used fingerprint algorithm increases [9].

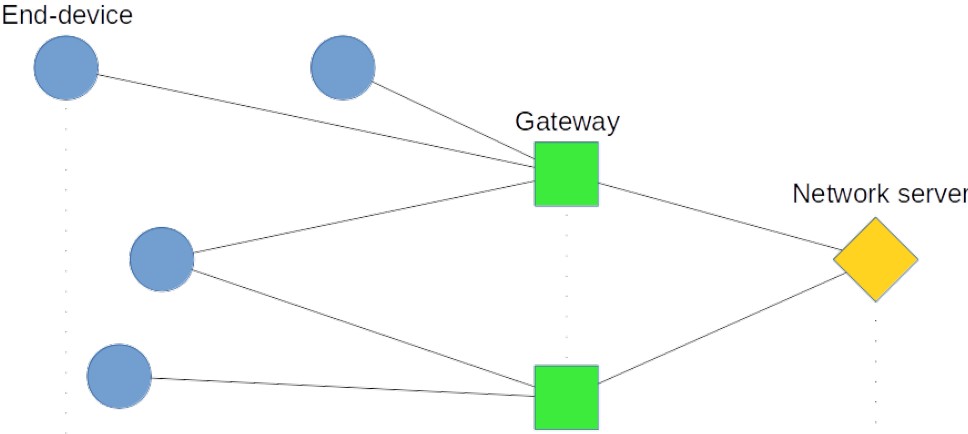

**Figure 1.** LoRaWAN architecture [8].

## 3. Related Work

Various indoor positioning models have been developed in the literature using different techniques, including range-based techniques such as Time of Arrival (ToA) [10], Angle

of Arrival (AoA) [11], and RSSI fingerprint matching algorithms [12]. Wireless technologies such as WiFi, Bluetooth, Radio Frequency Identification Device (RFID), UltraWideband (UWB), and LoRaWAN are considered by several researchers in developing indoor models. Poulose et al. in [13] used Deep Learning LSTM Networks to develop an indoor positioning system with UWB and achieved significant results. The authors in [14,15] used WiFi to develop an efficient indoor localization system. In addition, the authors in [16] did a survey and reported on the performance of different indoor localization systems and technologies. Sadowski et al. in [17] used the trilateration localization technique for indoor localization using LoRaWAN and obtained a minimum mean localization error of 1.19 m. Experimental data set evaluation of RSSI fingerprint-based indoor and outdoor localization networks was undertaken by the authors in [18,19].

Additionally, RNN have lately been applied to develop robust systems with significant results as reported by the authors in [20–22] for Heating, Ventilation, and Air Conditioning (HVAC) applications; [23] for non-occupied buildings' energy prediction; and [24–26] for intrusion detection applications. Moreover, according to our previous studies in [6,27], we developed LoRaWAN based localization systems on small and large-scale dense urban environments using RNN and achieved higher accuracy than the related work. Similarly, RNN is easily implemented on hardware compared to other algorithms in the literature because minimal instances are enough to represent RNN neurons [28,29]. Moreover, RNN predicts unobserved patterns not initially included in training data with a higher-level accuracy when compared to the performance of traditional Artificial Neural Networks (ANN) [30]. Furthermore, according to [31], RNN performed better than ANN in run-time even though at the expense of a higher training time, and RNN has a more robust generalisation capacity of the uncovered patterns during the training phase. Nonetheless, no one has explored RNN algorithms applied in ED positioning models for IoT location-aware applications in an indoor environment to date.

## 4. Methodology

This section describes the detailed procedure used to collect and pre-process the measurement data, mainly LoRaWAN RSSI values and ED position coordinates that we used to develop the proposed LoRaWAN-based indoor positioning system using RNN.

### 4.1. Dataset Collection Setup

We use MultiTech mDot conduit devices that are configurable and scalable Industrial LoRaWAN gateways [32], and a MultiTech mDot Box, a handheld unit with a fully functioning mobile ED containing a radio, and internal sensors for atmospheric pressure, altitude, temperature, light level, and accelerometer [33]. The MultiTech conduit gateway's server mode capability enables it to simulate the cloud-based system and acts as a stand-alone system working as a network server, and allows it to be easily used independently across a single site without an internet connection. Each conduit gateway is pre-configured with a static IP address and a web-based graphical user interface to control set up and logging using Java. To quickly access the graphical user interface, we directly connect a Lenovo laptop to the gateway ethernet port with a standard straight-through patch lead, and a static device IP is used to communicate with it initially on a windows machine for configuration. We also turn on another graphical user interface that can be run simultaneously as a control console on another web page generated by the conduit called Node-RED. This is to confirm the conduit is well configured, is receiving, and can process radio packets.

With the conduit and the ED now set up, the ED transmits RSSI, atmospheric pressure, altitude, temperature, light level, accelerometer, and signal information, among others, to the conduit gateways as a metadata message information provided by the network server written in JSON Javascript object notation format. Moreover, the packet also contains the LoRaWAN used Coding Rate (Codr = 4/5), Spreading Factor (SF = 12), Bandwidth (BW = 125 KHz), and the used frequency (Freq = 868 MHz). Furthermore, all the

recorded data are recovered later using mDot micro–Development Board with an 8-parity programmable cable using Tera Term software [34] on Windows.

### 4.2. Study Environments

We collect RSSI data at different X, Y ground truth grid points in both LOS and NLOS indoor environments in Destiny College main building in Glasgow city [35]. The three considered environments are 8 m × 22 m, 15 m × 25 m, and 20 m × 25 m meeting/lecture rooms on the ground floor (floor 0), floor 1, and floor 2, respectively, with varying propagation conditions. One of the two gateways is placed on a table in a corner on floor 0 and another on floor 2. The ED held by a moving user periodically sends a beacon message to the receiver gateways for two-minute transmissions at different grid position points from the three rooms on each of the three floors. The data collected from each environment are concatenated to form a dataset of 1205 and 1269 data samples in LOS and NLOS, respectively. A random sample picture of the data collection site is shown in Figure 2.

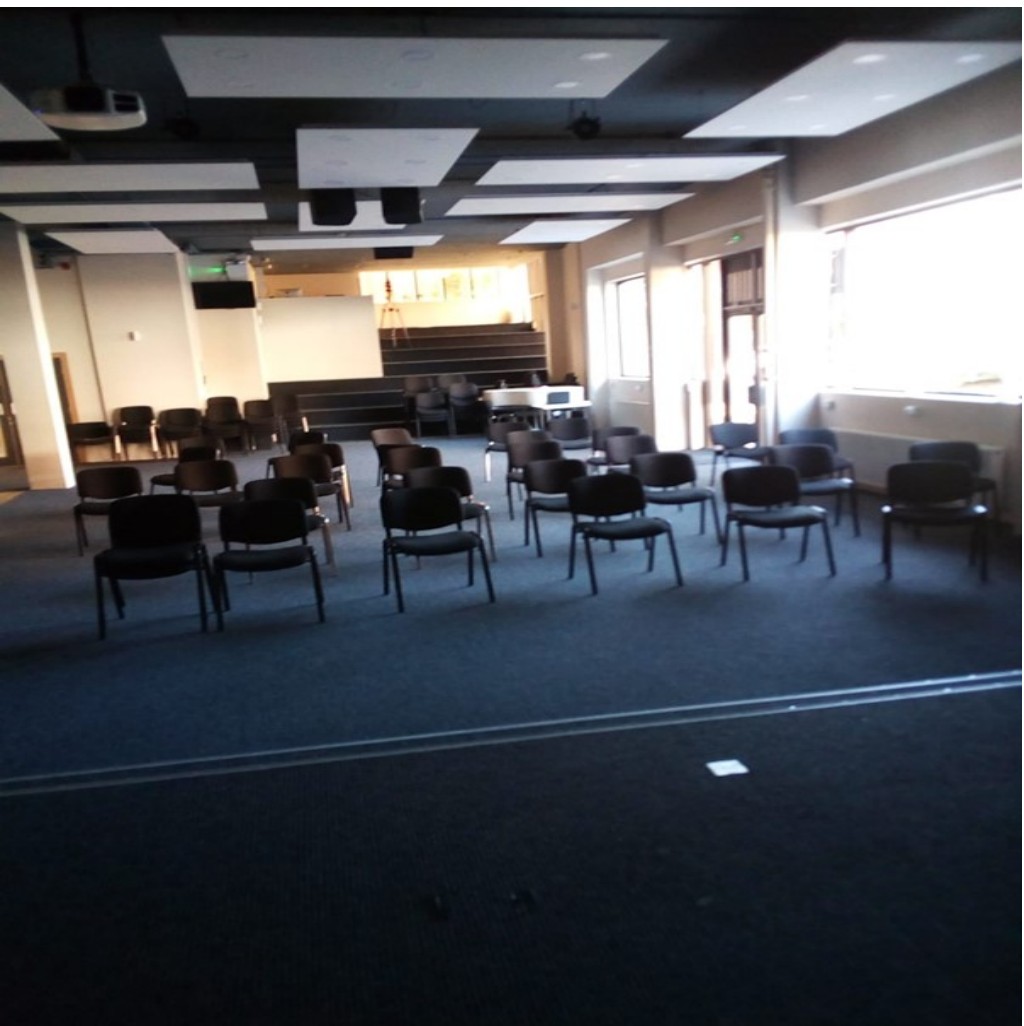

**Figure 2.** A random sample picture of the data collection site.

### 4.3. Data Normalization

We use the Min–Max Normalisation technique to scale our data in the range 0 to 1 for the large weights due to the large RSSI dataset, which causes the network to be unstable. The following formula is used:

$$y_i = \frac{RSSI_i - \min(RSSI)}{\max(RSSI) - \min(RSSI)}, \tag{1}$$

where $y(i)$ is the normalized data and $RSSI = (RSSI_1, ..., RSSI_n)$ is the raw RSSI input data.

### 4.4. Proposed LoRaWAN-Based Indoor Localization System Using RNN

RNN is a mathematical description of a novel category of artificial neural networks developed by Gelenbe [36]. In RNN, N number of connected neurons exchange positive and negative spiking impulse signals. A negative potential $(-1)$ represents inhibited signals, and a positive potential represents an excited signal to receiving neuron. A non-negative integer $K_i(t)$ represents the potential of every neuron $i$ at time $t$. If $K_i(t) > 0$, the neuron $i$ is in excited state and if $K_i(t) = 0$, $i$ is in idle state. The neuron $i$ transmits a signal to the receiving neuron $j$ at the Poison rate $r_i$ when excited. The signal transmitted to the next neuron either in excited or in the inhibited state with probabilities $p^+(i, j)$ or $p^-(i, j)$, respectively. Additionally, the transmitted signal information can depart the network with a probability described by the following mathematical formula:

$$c(i) + \sum_{j=1}^{N} p^+(i, j) + p^-(i, j) = 1, \forall i, \tag{2}$$

$$w^+(i, j) = r_i p^+ + (i, j) \geq 0, \tag{3}$$

Equally

$$w^-(i, j) = r_i p^- + (i, j) \geq 0. \tag{4}$$

Equations (2)–(4) combined

$$r(i) = (1 - c(i))^{-1} \sum_{j=1}^{N} [w^+(i, j) + w^-(i, j)] \tag{5}$$

The transmission rate for neurons in Equation (5) is $r(i)$, and is described as $r(i) = \sum_{j=1}^{N} [w^+(i, j) + w^-(i, j)]$. Furthermore, "$w$" represents the matrices of weight updates from neurons, and it is always positive as it is a product of transmission rates and probabilities.

An excited signal arrives at neuron $(i)$ with a positive potential, and is represented by Poison rate $\Lambda(i)$ and an inhibited signal with a negative signal at a Poisson rate $\lambda(i)$. Thus, the output activation function for each node "$i$" is represented by:

$$q(i) = \frac{\lambda^+(i)}{r(i) + \lambda^-(i)}, \tag{6}$$

whereby

$$\lambda^+(i) = \sum_{j=1}^{n} q(j)r(j)p^+(j, i) + \Lambda(i), \tag{7}$$

then

$$\lambda^-(i) = \sum_{j=1}^{n} q(j)r(j)p^-(j, i) + \lambda(i). \tag{8}$$

Gradient Descent (GD) is used to train the proposed LoRaWAN-RNN based localization system, and the calculated weights and biases are updated to the neurons as the algorithm computes the error. GD is a first-order iterative optimization algorithm generally

used by different researchers for training. GD minimizes the cost function, and the error cost function is presented by:

$$E_p = \frac{1}{2} \sum_{i=1}^{n} \gamma_i (q_j^p - q_j^p)^2, \gamma_i \geq 0 \tag{9}$$

where $\gamma \in (0,1)$ presents the status of the output neuron $i$, $q_j^p$ is a real differential function, and $q_j^p$ is the predicted output value. As per Equation (9), to find the local minima and minimize the error value of the error cost function, the relationship between neurons $y$ and $z$ is used, where weights $w^+(y,z)$ and $w^-(y,z)$ are updated by:

$$w_{y,z}^{+t} = w_{y,z}^{+(t-1)} - \eta \sum_{i=1}^{n} \gamma_i (q_j^p - y_j^p) [\frac{\partial q_i}{\partial w_{y,z}^+}]^{t-1}, \tag{10}$$

Furthermore:

$$w_{y,z}^{-t} = w_{y,z}^{-(t-1)} - \eta \sum_{i=1}^{n} \gamma_i (q_j^p - y_j^p) [\frac{\partial q_i}{\partial w_{y,z}^-}]^{t-1}. \tag{11}$$

Further details about GD and RNN are presented in [23].

RNN is used to model the developed system to map the input to the output using the collected LoRaWAN RSSI values and corresponding X, Y ground truth coordinates. The developed model is then used to predict any unknown X, Y position coordinates based on ED RSSI values, as shown in Figure 3.

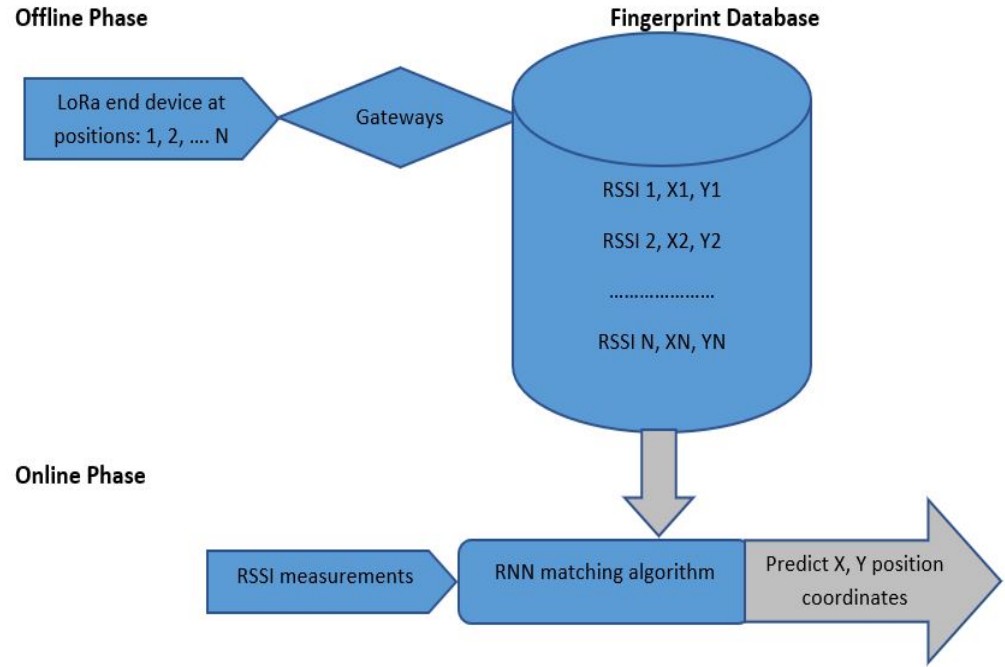

**Figure 3.** RNN based RSSI fingerprint localization method.

RNN is used to train and test the developed LoRaWAN based positioning system as a regression model with gradient descent algorithm. Various experiment setups are designed whereby 80% of 1205 and of 2269 collected data points in LOS and NLOS, respectively, are used to train the model, and 20% of the respective dataset is used to test the model. The developed RNN model considers four input neurons, 8, 16 and 20 hidden neurons, and two output neurons using 0.0002, 0.002, 0.2, and 2 learning rates in LOS and NLOS scenarios.

## 5. Results and Analysis

MATLAB R2021b is used to do extensive simulations to analyze the accuracy of the developed RNN-based localization model's accuracy in LOS and NLOS. The system's average localization error values obtained with 8, 16, and 20 numbers of hidden neurons, using 0.0002, 0.002, 0.02, 0.2, and 2 as learning rates are recorded. The following formula for Average Localization Error (AE) is used to analyse the accuracy and performance of the developed model:

$$AE = \sum_{i=1}^{n} ((X_{real} - X_{pred})^2 + (Y_{real} - Y_{pred})^2)^{0.5} \qquad (12)$$

Moreover, $(X_{real}, Y_{real})$ is the pre-recorded actual ground truth position coordinates, $(X_{pred}, Y_{pred})$ is the predicted position of unknown location estimated by the novel proposed localization model, and the total number of data points used is represented by n.

### 5.1. Results Analysis in LOS

From Figure 4, results show that the proposed RNN based indoor localization system's accuracy in LOS improves as the learning increases from 0.0002 to 0.2 with all the used network architectures. However, further increasing the learning rate to 2 increases localization error. The minimum AE obtained was 0.1219 m with an accuracy of 99.52% when 20 hidden neurons were used to model the proposed RNN network architecture with 0.2 as the learning rate. Furthermore, a training error of 0.06 m was obtained calculated in Root Mean Square (RMS), and the computational training time was 86.45 s. Table 1 presents AE values obtained for the used RNN network architectures with the used learning rates.

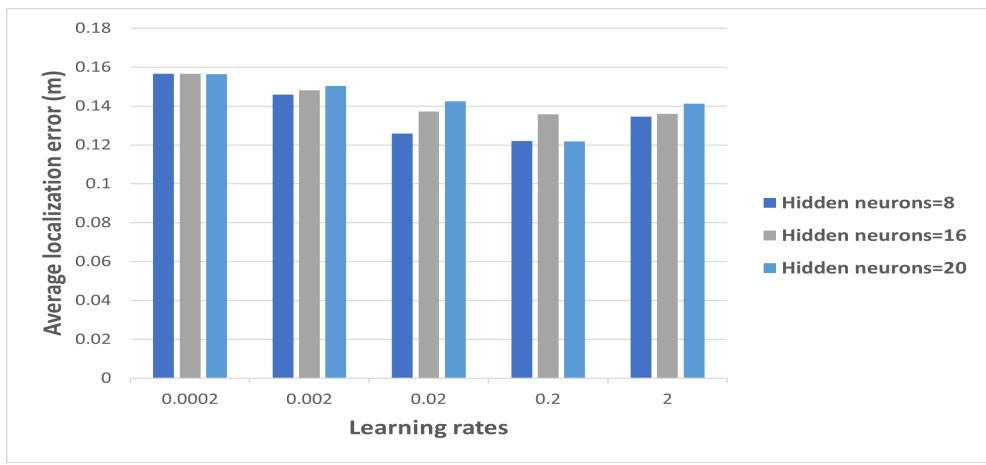

**Figure 4.** Localization accuracy of the used RNN network architectures with the considered learning rates in LOS.

**Table 1.** Accuracy of the used RNN based localization architectures with the used learning rates in LOS.

| Learning Rates | Average Localization Error (m) | | |
|---|---|---|---|
| | 8 Hidden Neurons | 16 Hidden Neurons | 20 Hidden Neurons |
| 0.0002 | 0.1567 | 0.1566 | 0.1565 |
| 0.002 | 0.1459 | 0.1481 | 0.1504 |
| 0.02 | 0.1258 | 0.1373 | 0.1424 |
| 0.2 | 0.1220 | 0.1358 | 0.1219 |
| 2 | 0.1345 | 0.136 | 0.1412 |

### 5.2. Results Analysis in NLOS

The proposed system's specific average localization error values obtained while using 8, 16, and 20 hidden neurons and 0.0002, 0.002, 0.02, 0.2, and 2 learning rates in the NLOS scenario are presented in Table 2. From Figure 5, the minimum average localization error is 13.94 m with an accuracy of 44.24% using a learning rate of 0.2 and 20 hidden neurons. The computational training time was 102.16 seconds with the minimum training MSE value of 0.09 m.

**Table 2.** Accuracy of the used RNN based localization architectures with the used learning rates in NLOS.

| Learning Rates | Average Localization Error (m) | | |
|---|---|---|---|
| | 8 Hidden Neurons | 16 Hidden Neurons | 20 Hidden Neurons |
| 0.0002 | 14.10 | 14.09 | 13.97 |
| 0.002 | 13.95 | 13.97 | 14.08 |
| 0.02 | 13.97 | 13.96 | 14.05 |
| 0.2 | 13.95 | 14.04 | 13.94 |
| 2 | 13.96 | 13.98 | 13.95 |

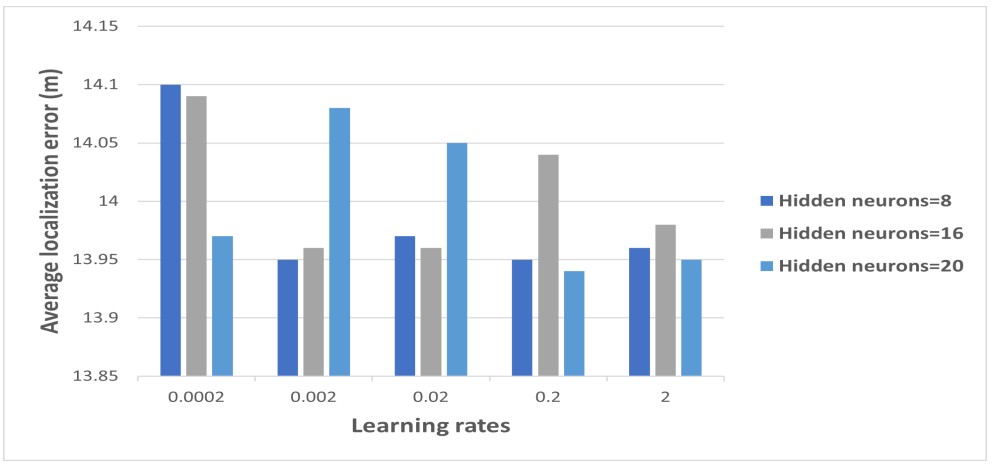

**Figure 5.** Localization accuracy of the used RNN network architectures with the used learning rates in NLOS.

### 5.3. Comparative Performance Analysis

The performance and position accuracy of LoRaWAN in an indoor environment both in LOS and NLOS scenarios is analyzed using the RNN algorithm to map position points to the corresponding RSSI values. The predicted distances are then compared to the ground-truth pre-recorded position coordinates to compute the localization error. The obtained results for the proposed localization system in LOS and NLOS scenarios are compared to the results obtained by different localization systems presented in the literature using polynomial regression, trilateration, path loss, time of arrival, and smoothing spline methods.

From Table 3, it can be seen that our developed LoRaWAN-RNN position system achieved the highest accuracy with the minimum average localization error of 0.12 m in LOS compared to 0.71 m presented in [37], 1.19 m in [17], 1.6 m in [38], 3.06 m in [7], and 8 m in [39], respectively. Moreover, the proposed system achieved 83.1%, 89.9%, 92.5%, 96.1%, and 98.5% improvement in average localization error compared with the results achieved by the authors in [7,17,37–39], respectively. The obtained results give important insights in developing LoRaWAN based indoor localization in LOS using RNN, and this can be better applicable in big indoor sports halls or fields, big hospital wards, shopping malls, and airports.

**Table 3.** RNN indoor localization results (LOS/NLOS) compared to results from related work.

| Research Study | AE (m)-LOS | AE (m)-NLOS | Method |
|---|---|---|---|
| Proposed localization system | 0.12 | 13.94 | RNN |
| Islam et al. [37] | 0.71 | 3.72 | Polynomial regression |
| Sadowski et al. [17] | 1.19 | - | Trilateration |
| Han et al. [40] | - | 1.8 | KNN |
| Kim et al. [38] | 1.6 | 3.1 | Trilateration |
| Anjum et al. [7] | 3.06 | - | Path loss |
| Henriksson [39] | 8 | - | Time of Arrival |
| Anjum et al. [41] | - | 9.38 | Smoothing spline |
| Manzoni et al. [42] | - | 20 | Trilateration |

Nevertheless, the proposed localization system's performance of the minimum average localization error of 13.94 m in NLOS was worse compared to 1.8 m in [40], 3.1 m in [38], and 9.38 m in [41], respectively. However, the developed model achieves higher accuracy compared to 20 m obtained by the authors in [42] with a 30.3% improvement in average localization error. This came due to the loss of LoRaWAN signal at different position points in NLOS, mostly when the ED and the receiving gateways were on different floors. Multi-path reflections are among the suggested potential causes of the poor performance in NLOS. However, the obtained results are proper for managing a fleet of vehicles within a pre-defined area in the NLOS.

## 6. Conclusions and Future Work

In this study, we propose a LoRaWAN- RNN based localization system using 0.0002, 0.002, 0.02, 0.2, and 2 as learning rates as well as 8, 16, and 20 hidden neurons to evaluate the performance and accuracy of the developed localization system in an indoor environment both in LOS and NLOS scenarios. A minimum average localization error of 0.12 m, which is 99.52% accurate, is achieved in LOS using a learning rate of 0.2, and 20 hidden neurons are used for the RNN network architecture. Moreover, the developed model outperforms the existing research studies in the literature with the minimum average localization error in LOS. The obtained results show that the developed LoRaWAN-based indoor localization model using RNN performs better than the existing models in LOS indoor scenarios and can be applied in large indoor sports halls or fields, big hospital wards, shopping malls, airports, and many more environments. Furthermore, a minimum localization error of 13.4 m with an accuracy of 44.24% in the NLOS was obtained, and this can be applied in intelligent car parking applications in indoor car parks whereby any vehicle can be located or routed within an industrial facility. In addition, it can also manage a fleet of vehicles within a defined area in NLOS indoor environment and send an alert if any vehicle is moved out of the pre-defined area, allowing control and avoiding theft. In our future work, we will use a new set of parameters for modelling the proposed system for an improved performance in NLOS. Furthermore, we will perform a comparative performance analysis of the developed system with other algorithms and evaluate the effects of multi-path reflections of LoRaWAN signals in a NLOS indoor environment.

**Author Contributions:** Writing—original draft, W.I.; and Writing—review and editing, H.L., R.M.G., and A.-U.-H.Q. All authors have read and agreed to the published version of the manuscript.

**Funding:** The Commonwealth Scholarships in the UK supported this work in partnership with the Government of Rwanda.

**Institutional Review Board Statement:** Not applicable.

**Informed Consent Statement:** Not applicable.

**Data Availability Statement:** Not applicable.

**Conflicts of Interest:** The authors declare no conflicts of interest.

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
