# Peer review of "LoRaWAN Based Indoor Localization Using Random Neural Networks"

_information, doi:10.3390/info13060303_

Round 1
Reviewer 1 Report
The Authors introduce the joint use of LoRa and RNN to manage the indoor localization problem.
The proposed technique performance in a LOS scenario is acceptable, but in a NLOS scenario the results are frustrating! The paper does not consider some important issues, such as the presence of reflections, multi-path effects, and so on.
No information is given regarding the accuracy of the measurements.
1 .The text must be thoroughly revised as it contains errors and typos (e.g.: abstract, 2nd sentence: ‘However, GPS do not work indoors due..’ do ---> does
2 .The proposed method efficiency cannot be evaluated, missing any evaluation of its computational time and a comparison with alternative numerical methods (RSSI fingerprint, MUSIC…)
3. Last sentence in the abstract and row 42,pg.2: 13.94m is quite a big error. If the dimensions of the experimental environments are those specified @ row 125, pg 4, the error is unbearable!
Please specify the error as a percentage of the distance
Row 22 pg 1 – Cost is not an issue, as we talk of a few euros per unit
Rows 70-72 – Add some references. I suggest:
doi: 10.1109/TAP.2012.2232893
10.1109/TCSI.2008.924126
doi: 10.1109/RADIO.2016.7772043
doi: 10.1109/TVT.2016.2545523
doi: 10.1109/TCSII.2020.2995064
doi: 10.23919/EuMC.2018.8541736
doi: 10.1109/NEWCAS.2017.8010174
Row 98, pg.3 and Figg.2-3 : being the so-called ‘set-up’ a commercially available device, the Figures are an unnecessary advertisement. The references in row 100-103 are sufficient
Pg.6 – Fig.4 seems quite unnecessary!
Author Response
CORRECTIONS FOR REVIEWER 1 COMMENTS FOR INFORMATION-1728158
Thank you so much for all your comments. The table below explains point by point, the details of the revisions to the manuscript and our responses to the comments.
Reviewer 1 Comments & Corrections
|
Comments |
Responses |
|
The proposed technique performance in a LOS scenario is acceptable, but in a NLOS scenario the results are frustrating! The paper does not consider some important issues, such as the presence of reflections, multi-path effects, and so on.
|
Multi-path reflections are among the suggested potential causes of the poor performance in NLOS (Now added in line 239-240). The effects of multi-path reflections of LoRaWAN signals in a NLOS indoor environment is a significant research point, which will be considered in our future works. |
|
No information is given regarding the accuracy of the measurements.
|
Percentage accuracy is presented in line 12, 15, 43-44, 203, 212, 247 and 254 |
|
1. The text must be thoroughly revised as it contains errors and typos (e.g.: abstract, 2nd sentence: ‘However, GPS do not work indoors due..’ do ---> does
|
The text has been revised to remove grammar errors and typos |
|
The proposed method efficiency cannot be evaluated, missing any evaluation of its computational time and a comparison with alternative numerical methods (RSSI fingerprint, MUSIC…)
|
The computation training time is now added in line 206 and 213 for both LOS and NLOS. A comparative performance analysis of the developed system with other algorithms is considered as our future work (added in line 260-261) as modelling using other algorithms is a significant research study not initially considered in this scope but set as our next study. |
|
Last sentence in the abstract and row 42, pg.2: 13.94m is quite a big error. If the dimensions of the experimental environments are those specified @ row 125, pg 4, the error is unbearable! Please specify the error as a percentage of the distance
|
Percentage accuracy presented in line 12, 15, 43-44, 203, 212, 247 and 254 |
|
Row 22 pg 1 – Cost is not an issue, as we talk of a few euros per unit
|
Now updated in line 22 |
|
Rows 70-72 – Add some references. I suggest: doi: 10.1109/TAP.2012.2232893 10.1109/TCSI.2008.924126 doi: 10.1109/RADIO.2016.7772043 doi: 10.1109/TVT.2016.2545523 doi: 10.1109/TCSII.2020.2995064 doi: 10.23919/EuMC.2018.8541736 doi: 10.1109/NEWCAS.2017.8010174
|
Relevant and recent references are added in the Related Work section |
|
Row 98, pg.3 and Figg.2-3 : being the so-called ‘set-up’ a commercially available device, the Figures are an unnecessary advertisement. The references in row 100-103 are sufficient
|
Figures 2-3, and line 98 are now removed |
|
Pg.6 – Fig.4 seems quite unnecessary!
|
Fig.4 removed |

Reviewer 2 Report
Authors propose a localization scheme based on random neural networks.
Although the proposed scheme performs better than related work in terms of AE-LOS, the paper needs to be revised as follows:
First, the paper is not well presented. The presentation of the paper needs to be improved.
Second, authors need to specify how the proposed scheme is distinct to other related work based neural networks.
Author Response
CORRECTIONS FOR REVIEWER 2 COMMENTS FOR INFORMATION-1728158
Thank you so much for all your comments. The table below explains point by point, the details of
the revisions to the manuscript and responses to the
comments.
Reviewer 2 Comments & Corrections
|
Comments |
Responses |
|
First, the paper is not well presented. The presentation of the paper needs to be improved.
|
The paper presentation is updated throughout. |
|
Second, authors need to specify how the proposed scheme is distinct to other related work based neural networks.
|
A brief description indicating the novelty of the neural network approach compared with current literature is now added in line 93-100 |

Round 2
Reviewer 1 Report
the paper has been revised according to the suggestions.